# HYPER-MULTI-STEP: THE TRUTH BEHIND DIFFICULT LONG-CONTEXT TASKS

## ABSTRACT

Long-context language models (LCLM), characterized by their extensive context window, is becoming increasingly popular. Meanwhile, many long-context benchmarks present challenging tasks that even the most advanced LCLMs struggle to complete. However, the underlying sources of various challenging long-context tasks have seldom been studied. To bridge this gap, we conduct experiments to indicate their difficulty stems primarily from two basic issues: "multi-matching retrieval," which requires the simultaneous retrieval of multiple items, and "logic-based retrieval," which necessitates logical judgment within retrieval criteria. These two problems, while seemingly straightforward, actually exceed the capabilities of LCLMs because they are proven to be hyper-multi-step (demanding numerous steps to solve) in nature. This finding could explain why LLMs struggle with more advanced long-context tasks, providing a more accurate perspective for rethinking solutions for them.

## 1 INTRODUCTION

In the past year, long-context language models (LCLMs) such as GPT-4o-128k (OpenAI, 2023) and Gemini-1.5-1000k (Team et al., 2023) have surged in popularity, raising questions about their efficacy in handling extended context tasks. While various LCLMs have demonstrated excellent long-context retrieval ability by passing the "Needle in a Haystack" test (gkamradt, 2023) in over 100k context length, benchmarks like Loogle (Li et al., 2023) and Loong (Wang et al., 2024b) have highlighted their shortcomings in more complex tasks.

To better emulate real-world challenges, recent long-context benchmarks predominantly aim to enhance the difficulty of long-context tasks by requiring real-life scenarios understanding and multi-step processes or increasing the volume of information that needs to be aggregated. This lead these tasks to usually engage multiple capabilities of LLMs, such as world knowledge, advanced reasoning skills and retrieval capability. Consequently, poor model performances are often vaguely attributed to inadequate understanding, reasoning and retrieval capabilities in long context, making it challenging to identify the specific factors that constitute the primary difficulty.

To reveal the real source of challenges in long context tasks, we conduct a detailed analysis (detailed in Appendix C.2 and C.3) of challenging tasks from previous long-context benchmarks, and as expected, we identify 2 common factors that make them difficult: multi-matching retrieval and logic-based retrieval. Multi-matching retrieval involves recalling multiple items simultaneously, and logic-based retrieval involves logical judgment within retrieval criteria. Although they are both "basic" retrieval problems having a straightforward form and cannot be explicitly decomposed into multiple steps using Chain-of-Thought (Wei et al., 2022) (in Appendix C.1, we use examples to detail their differences from those traditional multi-step tasks which can be decomposed by CoT, hence called "formally multi-step"), our experiments, as shown in Figure 1, demonstrate they are much harder for current LCLMs as the context length grows, compared to direct retrieval or formally multi-step retrieval.

Rather than merely focusing on the superficial phenomena, we endeavor to elucidate why these ostensibly simple issues present substantial challenges for LLMs. Through more in-depth experiments, we demonstrate that they are "hyper-multi-step" in nature, which are quite distinct from normal retrieval problems. "Hyper-multi-step", the truth behind difficult long-context tasks, refers to a problem that appears indivisible in form but actually requires numerous independent steps,

and the number of steps will increase indefinitely with the length of the context, that exceed the capacity of LLMs to process simultaneously. To date, none of the techniques such as Retrieval-Augmented Generation (RAG), Chain-of-Thought (CoT) prompting and LCLMs have adequately addressed such problems.

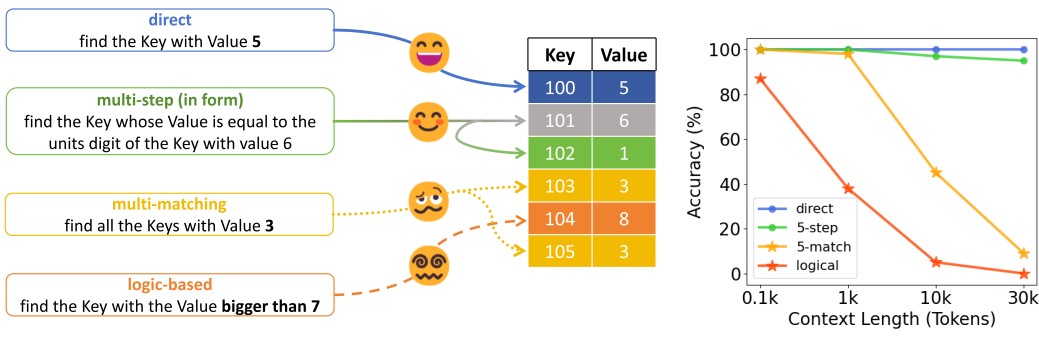

(a) different retrieval problems        (b) accuracy of retrieval problems

Figure 1: (a) Examples of direct, formally multi-step, multi-matching and logic-based Key-Value retrieval. (b) Accuracy of GPT-4o in different KV retrieval tasks, as the context length increases. 5-step means a multi-step retrieval task requiring at least 5 times of retrieval. 5-match means a multi-matching retrieval task with 5 matching items for one query. Logical means logic-base retrieval.

Through our studies, we do not aim to propose more challenging benchmarks to drive further optimizations in LCLMs. Rather, we demonstrate a tough reality: while LCLMs are intended to process vast amounts of input data simultaneously, there exist certain long-context tasks which always remain unattainable for LCLMs to solve within one step. Therefore, future research should focus on addressing the challenges associated with numerous steps, rather than merely extending the context window of LLMs.

Our major contributions are as follows:

- We summarize previous long-context benchmarks and evaluate current LCLMs to identify 2 common factors that greatly contribute to the difficulty of long-context tasks: multi-matching retrieval and logic-based retrieval.

- We propose that the essence of the 2 problems is hyper-multi-step, and we provide detailed proof and explanations for this assertion. This was never specified in previous researches.

- We prove that LCLMs inherently have limitations, offering new insights for understanding and addressing long-context problems.

## 2 RELATED WORKS

### 2.1 LONG-CONTEXT LANGUAGE MODELS

The emergence of long-context language models (LCLMs) aims to enable language models to handle vast amounts of input information simultaneously. In recent years, closed-source LLMs have pioneered advancements in long-context modeling, with context windows expanding from 128k to 1000k tokens. Notable models include GPT-4o (OpenAI, 2023), Claude3.5-200k (Anthropic, 2024), and Gemini-1.5-1000k (Team et al., 2023), which are capable of processing significantly longer texts. Concurrently, open-source models such as phi-3.5-mini (Abdin et al., 2024) and Qwen2.5 (Team, 2024) leverage advanced RoPE (Su et al., 2021) interpolation techniques like Yarn (Peng et al., 2023) and LongRope (Ding et al., 2024) to achieve a 128k context window. These open-source models are usually extended from a 4k pretraining length through long-context post-training with interpolated RoPE. However, it remains to be seen whether these models can truly achieve accurate and efficient handling of lengthy contexts.

## 2.2 Long-Context Benchmarks

The field of long-context modeling is experiencing rapid advancements, with benchmarks evolving to become more complex and multifaceted. Simple synthetic tasks such as Needle-in-a-Haystack (NIAH) (gkamradt, 2023) assess the retrieval abilities of long-context language models. Earlier benchmarks such as Longbench (Bai et al., 2023), BAMBOO (Dong et al., 2024), and L-eval (An et al., 2023) offer a wide-ranging evaluation of long-context comprehension through diverse task forms, though they typically lack an emphasis on difficulty. More recent benchmarks, including InfiniteBench (Zhang et al., 2024), RULER (Hsieh et al., 2024), LOOGLE (Li et al., 2023), and LOONG (Wang et al., 2024b), incorporate harder tasks with varying levels of complexity and adjustable context length. Meanwhile, LOFT (Lee et al., 2024) examines whether long-context models can function as retrieval systems like RAG and SQL. Despite these advancements, few studies have thoroughly investigated the underlying commonalities of these intricate long-context tasks, leaving a gap in understanding the fundamental reasons behind their challenges.

## 3 MULTI-MATCHING AND LOGIC-BASED RETRIEVAL ARE REALLY HARD

In this section, we conduct evaluations on current LCLMs to prove that, multi-matching and logic-based retrieval are really hard, nearly unsolvable for current LCLMs in long context. In contrast, normal retrieval or formally multi-step retrieval, proves to be much easier (as illustrated in Appendix B.1 and B.2).

### 3.1 EXPERIMENTAL SETTINGS FOR MODEL EVALUATION

To prevent data contamination, we create two fully synthetic datasets: Key-Value Pair Retrieval and Student Resume Retrieval. These datasets make it easy for us to make controlled changes to the input context size and problem types.

In Key-Value pair retrieval, the context is a JSON-formatted dictionary consisting of $N$ randomly generated Key-Value pairs. The Key is a 10-digit string, and the Value is a positive integer. The question is appended to the context and varies based on the task type. For multi-matching, the model must retrieve all Keys associated with a given Value. For logic-based retrieval, the model needs to identify the Key with the Value within a specified range.

In the Student Resume Retrieval task, all original resumes are fictional and generated by GPT-4o. The context includes $N$ rows, each detailing a fictional college student's information, such as name, age, graduation school, interests, GPA, and a short self-introduction. In multi-matching problems, the task is to retrieve all students graduating from a specified university. For logic-based retrieval, the problem is to identify the student whose GPA falls within a specified range. All GPAs are rounded to two decimal places and range from 0 to 5. Examples of the prompts are as follows:

---

**Context (KV Retrieval)**

JSON data with 3000 Key-Value pairs:
{"1532968704": 78, "5921306748": 84, "3742815096": 47, ......, "3276918540": 76}

---

**Question (KV Retrieval)**

**Direct retrieval:** In the above JSON data, please find the value of the key '6978024153'. Provide your final answer in the format "value: {answer}".

- - - - - - - - - - - - - - - - - - - - - - - - - - - - - - - - - - - - - - - - - - - -

**Multi-matching retrieval:** In the above JSON data, please find all the keys with the value 0. Provide your answer (keys separated by commas) in the format "keys: {answer}".

- - - - - - - - - - - - - - - - - - - - - - - - - - - - - - - - - - - - - - - - - - - -

**Logic-based retrieval:** In the above JSON data, please find the key (only one) whose value is greater than 223 and smaller than 278. Provide your answer in the format "key: {answer}".

---

> **Context (Student Resume Retrieval)**
>
> Here are 100 students' resumes:
> * The student named Hallie Turner is 21 years old, graduated from New York University with a GPA of 4.96. He/She is interested in Social Media, Writing, and his/her self-introduction is: Creative writer exploring the impact of social media on culture.
> * The student named Sonali Jain is 22 years old, graduated from ......

> **Question (Student Resume Retrieval)**
>
> **Direct retrieval:** What is the age of the student named Hallie Turner?
> - - - - - - - - - - - - - - - - - - - - - - - - - - - - - - - - - - - - - - - -
> **Multi-matching retrieval:** Please find all the students who graduated from Tokyo University of Agriculture and Technology. Provide your final answer (students' names separated by commas) in the format "names: {answer}".
> - - - - - - - - - - - - - - - - - - - - - - - - - - - - - - - - - - - - - - - -
> **Logic-based retrieval:** Which student has a GPA between 2.25 and 2.58? Provide your final answer (student's name) in the format "name: {answer}".

We denote $N$ as the total number of items (an item corresponds to either a key-value pair or a student) in the input context and $n$ as the number of items to be retrieved. In KV retrieval, $N$ is set to 4, 10, 100, 1000 and the corresponding context length is 0.04k, 0.1k, 1k and 10k tokens respectively. In student resume retrieval, $N$ is set to 4, 10 and 100 and the corresponding context length is 0.3k, 0.6k and 6k respectively. For logic-based retrieval, $n$ is set to 1, and the model is informed that only one correct item exists. In multi-matching ($n$-matching) retrieval, $n$ is set to different values from 1 to 20, and the model is not informed of the number of matching items. The gold items (i.e. the items to be retrieved) are distributed at random positions within the context. For each problem setting, we construct 200 test samples with different context.

For direct or logic-based retrieval where $n = 1$, each question has the exact answer, making evaluation straightforward and objective. We use exact-match evaluation and accuracy as the metric for all tasks. For multi-matching retrieval, the model may generate a list of items as the answer, which may be partially correct and random in order. To calculate accuracy, we only consider the prediction correct if it is an totally exact match (without considering the order of the items) to the reference set, i.e., both over-selection and under-selection are considered incorrect.

We have experimented with different context formats, such as declarative sentences, JSON dictionary, and Markdown tables, as well as placing the question before the context. These variations show no significant impact on performance. Moreover, although logic-based retrieval can be more advanced forms, we restrict it to basic forms such as numeric comparison for simplicity. Therefore, we standardize the format of all prompts to align with the examples provided above across all experiments.

We evaluate 5 popular long-context models with context windows over 128k tokens, including Phi-3.5-mini-instruct (phi-3.5) (Abdin et al., 2024), Meta-Llama-3.1-70B-Instruct (llama3.1-70b) (Dubey et al., 2024), Deepseek-V2.5 (deepseek) (DeepSeek-AI et al., 2024), Gemini-1.5-flash (gemini) (Team et al., 2023), and GPT-4o-2024-08-06 (gpt-4o) (OpenAI, 2023). In all experiments, we set the temperature to 0.8, top_p to 0.9, and max generated tokens to 512.

## 3.2 MODEL PERFORMANCE ON MULTI-MATCHING RETRIEVAL

The results presented in Table 1 reveal that when only 1 matching item is present, larger models, such as Gemini (Team et al., 2023), demonstrate superior performance, achieving an accuracy of up to 94% even in lengthy contexts. However, with the introduction of multiple matching items, such as 5 or 10, the accuracy of all language models rapidly declines to nearly zero, particularly evident in the more realistic scenario of Student Resume Retrieval. This trend suggests that the inherent difficulty of the task is consistently challenging across models of varying sizes.

Additionally, we explore the use of Chain of Thought (CoT) prompts (Wei et al., 2022) by incorporating the phrase "let's think step by step." However, this approach yields minimal improvement,

| Model | Num Matches ($n$) | KV Retrieval | | | Student Resume Retrieval | |
|---|---|---|---|---|---|---|
| | | $N$=10 | $N$=100 | $N$=1000 | $N$=10 | $N$=100 |
| phi-3.5 | 1 | 91 | 42 | 1 | 98 | 33 |
| | 5 | 36 | 0 | 1 | 27 | 0 |
| | 10 | / | 0 | 0 | / | 0 |
| | 20 | / | 0 | 0 | / | 0 |
| llama3.1 70b | 1 | 100 | 99 | 62 | 100 | 98 |
| | 5 | 99 | 59 | 3 | 99 | 21 |
| | 10 | / | 45 | 0 | / | 0 |
| | 20 | / | 24 | 0 | / | 0 |
| deepseek | 1 | 100 | 95 | 0 | 100 | 100 |
| | 5 | 100 | 40 | 0 | 91 | 23 |
| | 10 | / | 13 | 0 | / | 0 |
| | 20 | / | 3 | 0 | / | 0 |
| gemini | 1 | 100 | 100 | 94 | 100 | 96 |
| | 5 | 100 | 82 | 66 | 100 | 36 |
| | 10 | / | 48 | 18 | / | 8 |
| | 20 | / | 35 | 2 | / | 1 |
| gpt-4o | 1 | 100 | 100 | 60 | 100 | 100 |
| | 5 | 100 | 98 | 45 | 100 | 65 |
| | 10 | / | 85 | 5 | / | 0 |
| | 20 | / | 50 | 0 | / | 0 |

Table 1: Accuracy (%) of two tasks: KV Retrieval and Student Resume Retrieval, requiring retrieving all matches under varying $N$. We also count different types of errors and detail the ratio of over-selection, under-selection and mis-selection in Appendix B.3.

likely because the problem is inherently simple in nature, preventing the language model from effectively decomposing it into actionable steps.

### 3.3 MODEL PERFORMANCE ON LOGIC-BASED RETRIEVAL

| Model | KV Retrieval | | | | Student Resume Retrieval | | |
|---|---|---|---|---|---|---|---|
| | $N$=4 | $N$=10 | $N$=100 | $N$=1000 | $N$=4 | $N$=10 | $N$=100 |
| phi-3.5 | 69 | 9 | 0 | 0 | 75 | 53 | 9 |
| llama-3.1-70b | 72 | 41 | 6 | 1 | 81 | 63 | 13 |
| deepseek | 84 | 67 | 12 | 0 | 94 | 81 | 21 |
| gemini | 78 | 33 | 6 | 0 | 92 | 80 | 12 |
| gpt-4o | 97 | 87 | 38 | 5 | 100 | 92 | 30 |

Table 2: The accuracy (%) on logic-based KV retrieval and Student Resume Retrieval.

As illustrated in Table 2, in logic-based retrieval, when the context contains only 4 options, LLMs successfully select the correct item in most cases, indicating that these models possess logical judgment capabilities. However, for both datasets, all tested models struggle with these tasks as the context length increases, consistently retrieving an incorrect item whose value lies outside the specified range. Similarly, the use of CoT (Wei et al., 2022) prompting does not mitigate this issue for the same reasons. It is reasonable to anticipate that these difficulties will be further exacerbated in more complex scenarios requiring logical judgment beyond mere numeric comparison.

# 4 MULTI-MATCHING AND LOGIC-BASED RETRIEVAL ARE HYPER-MULTI-STEP

To delve deeper into why LLMs struggle to solve these seemingly simple tasks, we aim to answer the following questions: Do these issues fundamentally differ from traditional retrieval at the model's internal level? Can these issues be further decomposed into simple solvable components? Can they be resolved with additional reasoning steps in inference-time?

Eventually, our findings reveal a consistent pattern: the number of steps required for these problems exceeds the limits of what LLMs can handle. Hence we refer to them as "hyper-multi-step", characterized by the accumulation of numerous simple steps, analogous to simultaneously solving thousands of elementary mathematics problems.

Specifically, our findings are as follows, which will be detailed in the following sections:

For logic-based retrieval:

1. The internal behavior of LLM in logic-based retrieval is more akin to arithmetic tasks (i.e. logical tasks) which necessitate reasoning with multiple steps.

2. While logic-based retrieval can be decompose to 2 components: decide which value is in the range, and then get the corresponding key, vector retrieval cannot even solve the first one within one step.

3. If we allow LLM to take $N$ steps in test-time, it can successfully accomplish logic-based retrieval.

And for multi-matching retrieval:

1. The model's internal behavior in multi-matching retrieval is originally an one-by-one retrieval process.

2. Even when we decompose a multi-matching retrieval problem into $n$ single-item retrievals, the difficulty of retrieval continuously increases for items searched later.

3. Even if LLM is allowed to take $N$ steps in test-time, the accuracy still falls short of perfection.

## 4.1 EXPERIMENT SETTINGS FOR HIDDEN STATES AND ATTENTION ANALYSIS

In our experiments, analyzing the internal behavior of the model is the most crucial. We employ linear probing to explore the information encapsulated in hidden states and statistically analyze attention weights to infer the model's internal mechanisms. Given our earlier observations of similar performance trends between large and small models, we adopt the lightweight small model, phi-3.5-mini (Abdin et al., 2024), which consists of 32 layers and has a hidden size of 3072, for simplicity.

The dataset used is still Key-Value retrieval, but more simplified: the total number of KVs does not exceed 100, the value range is constrained to 0 to 9, and we do not apply the chat template to allow the model to generate answers directly following the question.

In both hidden state and attention analyses, we first identify the anchor token, whose hidden states are used for linear probing and function as the query token in the attention mechanism. In tasks where $n$ is 1, the last token of the question serves as the anchor token. In multi-matching retrieval tasks, $n$ is set to 3, and the 3 gold Keys are appended to the question. The token immediately preceding each of the 3 appended Keys acts as the anchor token, i.e., we conduct experiments on 3 anchor tokens separately. We use examples to show where the anchor token is in Appendix A.7.

In examining hidden states, we employ linear probing to determine whether the model has successfully retrieved the correct Key and stored its information in the hidden states output by each layer. The label is the first digit of the gold Key to be retrieved, enabling our linear prober to function as a 10-class classifier with a single linear layer. We use 1,600 samples for training and 400 for testing the classifier, and for each layer, we train and test the classifier independently. We train it for 8 epochs with the learning rate of $10^{-5}$.

To elucidate the model's attention dynamics, we compute "relative attention" which the anchor token pays to the gold Key and Value in each layer. "Relative attention" is the ratio of the average attention weight directed towards the gold Key (or Value) to that towards all other candidate Keys (or Values), reflecting the model's capacity to focus on the target one and exclude distractors. The absolute attention weights are shown in Appendix B.4.

## 4.2 LOGIC-BASED RETRIEVAL

### 4.2.1 INTERNAL MODEL BEHAVIOR: MORE LIKE MULTI-STEP ARITHMETIC

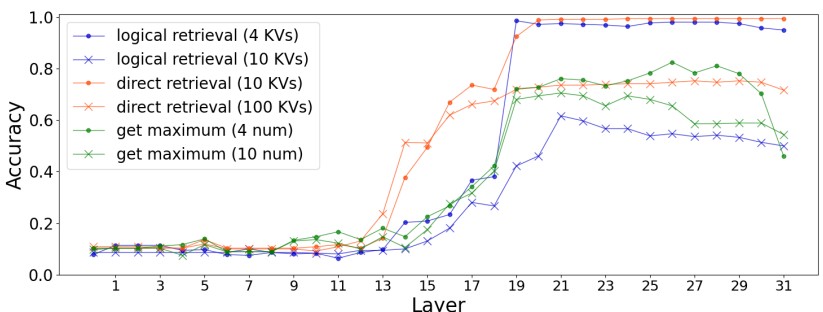

Figure 2: Linear probing accuracy in each layer of 3 tasks: direct KV retrieval, logical KV retrieval, and getting the maximum value among $N$ numbers.

First, we analyze the process of a logic-based KV retrieval task, comparing it with two other tasks: (1) direct KV retrieval, where the model retrieves the Key corresponding to a given Value, and (2) a basic arithmetic task that involves identifying the largest number among $N$ integers ranging from 0 to 100 (see prompts in Appendix A.5). In task (2), we use the ground-truth's unit's digit as the label for probing, and the last token of the prompt as the anchor token.

From the linear probing accuracy (Figure 2) and the attention dynamics (Figure 3), we find that model behavior in logic-based retrieval is fundamentally different from standard retrieval and more resembles the numeric comparison task:

1. For direct retrieval, probing accuracy increases in discrete jumps, notably between layers 14 and 20, rather than progressively across all layers. The relative attention toward the gold Key and Value also peaks between layers 15 and 23, suggesting that this attention correlates with the retrieval behavior. This indicates that retrieval activity is concentrated within particular layers (e.g. layers 14 and 20).

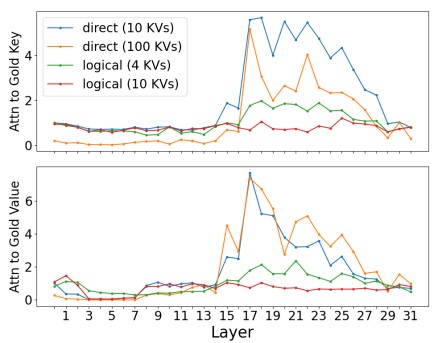

Figure 3: Relative attention to the gold Key and Value of each layer of different KV retrieval tasks.

2. In logic-based retrieval, the attention curves show a distinct trend and remain consistently lower than direct retrieval. What's more, the point at which the accuracy of logic-based retrieval begins to improve, layer 19, is noticeably later than that of direct retrieval, layer 14; instead, it more closely mirrors the numeric comparing task, which also begins to increase at layer 19. This indicates that logic-based retrieval problems differ fundamentally from normal retrieval tasks, bearing more resemblance to arithmetic problems.

A prior research (Feng et al., 2023) has theoretically demonstrated that a constant-size transformer model cannot solve arithmetic problems with many steps in a single step unless using CoT (Wei et al., 2022). Therefore, a logic-based retrieval, as it resembles arithmetic, will also be unsolvable in one step as long as $N$ is large enough.

### 4.2.2 CAN VECTOR RETRIEVAL SOLVE LOGIC-BASED RETRIEVAL?

Second, we intuitively decompose a logic-based retrieval problem into 2 components: decide which value is in the range, and get the corresponding key. Since our focus is on retrieval scenarios, we directly employ models designed for RAG (Lewis et al., 2020) to test the feasibility of performing the first step through vector retrieval. (The second step is the same as normal retrieval.)

Vector retrieval is one of the most widely retrieval techniques, which encodes both the query and candidates into embedding vectors, then calculates the similarity between the query and each candidate to retrieve the top similar options. This process is similar to the attention mechanism within LLMs (Xu et al., 2023). Since similarity computation can be conducted in parallel for each candidate, it can be considered as a single-step process from an external perspective.

We test 2 sentence embedding models commonly used for RAG (Lewis et al., 2020), e5-large-multilingual (Wang et al., 2024a) and bge-m3(Zhang et al., 2023), on a numerical value comparing task to see if it can retrieve the correct number from 20 integers. The candidate keys are integers within the range 0 to 30, 100, 1,000, or 10,000, and the queries have 2 types, equality relations and greater&less-than comparison (examples are shown in Appendix A.6). If the embedding vector of the correct number has the highest similarity to that of the query, it is considered correct. As shown in Table 3, they only function properly when retrieval is based on equality relations; however, their performance significantly declines with more advanced logical operations, such as greater-than or less-than comparisons.

This suggests that one-step vector retrieval techniques are inadequate for logic-based retrieval. In other words, a transformer model cannot achieve logic-based retrieval through the attention mechanism within a single layer (or a few layers); instead, it requires a more advanced reasoning process.

| Model | Criteria Type | within 30 | within 100 | within 1k | within 10k |
|-------|---------------|-----------|------------|-----------|------------|
| e5-large | Greater/Less Than | 36 | 31 | 21 | 16 |
| | Equal | 95 | 98 | 99 | 99 |
| bge-m3 | Greater/Less Than | 37 | 29 | 20 | 23 |
| | Equal | 95 | 98 | 99 | 100 |

Table 3: Accuracy of sentence embedding models on numerical comparison retrieval, with 2 different types of retrieval criteria, as the range of random selected numbers increases. The number of test samples are 100.

### 4.2.3 WHAT IF INVESTING N TIMES MORE TIME?

| Prompt | KV Retrieval | | Student Resume Retrieval | | Output Tokens |
|--------|--------------|----------------|--------------------------|----------------|---------------|
| | logic-based | multi-matching | logic-based | multi-matching | |
| standard | 38 | 50 | 13 | 0 | 12 |
| step-by-step | 47 | 52 | 37 | 10 | 340 |
| one-by-one | 100 | 90 | 100 | 20 | 1500 |

Table 4: Performance of GPT-4o if we prompt it to think step bu step or examine each item one by one. In all the experiments, $N$ is set to 100. In multi-matching retrieval, $n$ is set to 20 in KV retrieval and 10 in Student Resume Retrieval. In logic-based retrieval, $n$ is always 1.

Third, we find LLM can indeed better solve this problem if using an unconventional CoT-like method, paying as many steps as $N$, i.e. using an algorithm with a time complexity of $O(N)$. In experiments, we employ 3 types of prompts: (1) standard prompt, the default prompt without CoT; (2) step-by-step prompt, which is the traditional CoT prompt which tells the model to "think step by step, but do not check each item one by one"; (3) one-by-one prompt, which tells the model "You should first examine every item one by one to give the judgement (yes/no) on whether it meet the requirement".

As shown in Table 4, we find the accuracy is greatly improved to 100% with one-by-one prompt, though it costs hundreds of times more time (examples are shown in Appendix A.3). In contrast, traditional CoT prompt with limited output length cannot improve much. This indicates that if decomposed to enough steps, the hyper-multi-step problem can be solved in theory, but the time required is excessively long.

### 4.3 MULTI-MATCHING RETRIEVAL

#### 4.3.1 INTERNAL MODEL BEHAVIOR: RETRIEVE ONE BY ONE BUT INCREASINGLY HARD FOR RETRIEVING LATER ITEMS

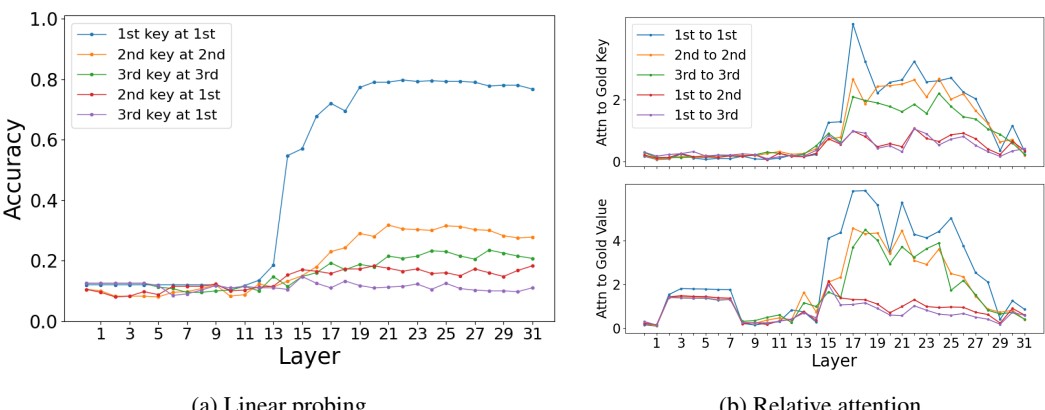

(a) Linear probing  (b) Relative attention

Figure 4: (a) The accuracy of probing from hidden states of each layer when predicting the first, second, and third Key. For example, "2nd key at 1st" means we probe from the hidden states of the 1st anchor token but use the 2nd anchor token (the 2nd gold Key's first digit) as the label for training and testing. (b) Attention to each Key or Value when predicting the first, second, and third Key respectively. For example, "1st to 2nd" means we calculate the attention to the 2nd gold Key (or Value) with the 1st anchor token as the query token.

For the multi-matching situation, first, we examine the model's behavior when tasked with predicting each matching item. The relative attention curve (Figure 4b) and the linear probing accuracy (Figure 4a) demonstrate that:

1. In multi-matching retrieval, the model retrieves the matching items one by one rather than simultaneously, as no information about the 2nd or 3rd key can be detected at the end of the question (i.e. the first anchor token), which proves the model does not retrieve multiple items all at once. Correspondingly, the attention to the 2nd or 3rd key at the end of the question is very low, and the model mainly focus on only the 1st key.

2. The model can accurately retrieve the first Key, akin to the behavior in direct retrieval. However, for the subsequent Keys, both attention and probing accuracy are much lower, with the 3rd Key being harder to retrieve than the 2nd. This proves predicting later items is increasingly harder for LLMs, despite there in fact being no priority relationship among the items.

#### 4.3.2 IS RETRIEVING JUST ONE ITEM SIMPLE?

Second, to further prove that retrieving a later item is indeed more challenging, we design a simplified version of the multi-matching Key-Value (KV) retrieval problem: the model is provided with all but one of the $n$ matching Keys and is required to predict the last remaining Key (see detailed prompts in Appendix A.1). The Key to be retrieved is selected randomly, meaning it may not necessarily be the last one that appears in the sequence of the input context.

The results in Table 5 demonstrate that even when the task is ostensibly reduced to predicting a single item, corresponding to just one step in the one-by-one retrieval process, the difficulty still increases to very high as the number of matching items grows. The results indicate that retrieving

| Model | Total KVs ($N$) | $n=3$ | $n=10$ | $n=30$ | $n=100$ |
|---|---|---|---|---|---|
| gpt-4o | 100 | 100 | 92 | 42 | / |
|  | 1000 | 95 | 42 | 25 | 1 |
| gemini | 100 | 100 | 99 | 67 | / |
|  | 1000 | 99 | 90 | 49 | 8 |

Table 5: Accuracy on KV retrieval, requiring retrieving only the last Key (of $n$ Keys) with the given Value, and other Keys with the given Value are already given. The results of other models are in Appendix B.5

even a single item is not simple, if the item is a later (the order in which items appear in the output sequence, rather than the input sequence) retrieved one.

We speculate the increasing difficulty may stem from the increasing complexity of retrieval criteria for later items. Retrieving subsequent Keys requires more extensive and stringent criteria, including conditions to exclude previously retrieved items, thereby complicating the retrieval process. In other words, the model need to pay more efforts to exclude previously retrieved items to avoid getting too many items at the same time. Therefore, multi-matching retrieval also has the multi-step nature: it not only needs $n$ steps to generate $n$ items for $n$-matching retrieval, and but also may need additional $k$-1 steps to exclude previous ones for the $k$-th item. Consequently, the number of steps required may be proportional to $n^2$.

### 4.3.3 WHAT IF INVESTING N TIMES MORE TIME?

Third, to demonstrate that the complexity of solving multi-matching is even more than $O(N)$, we conducted similar experiments to prompt the model to examine every item one by one. The results shown in Table 4 indicate that investing time proportional to $N$ can indeed improve accuracy compared to one-step solutions, but still cannot ensure perfect correctness. However, though we already prove it is multi-step, we remain uncertain about its exact complexity level, because the current LLMs perform so poorly on multi-matching tasks, that we have not identified an appropriate prompt or an effective reasoning process to let LLM adequately address this issue.

### 4.4 POTENTIAL SOLUTIONS FOR HYPER-MULTI-STEP PROBLEMS

As we have found, multi-matching and logic-based retrieval problems are hyper-multi-step, meaning the number of steps required exceeds the limit that current LLMs can directly handle. Therefore, relying solely on the LLM itself provide the complete answers is not only inefficient but also very error-prone.

Nevertheless, similar to how humans tackle such problems, using external tools may provide a viable solution. For example, if the input context is well-structured (e.g. Markdown tables or JSON data), the model can be prompted to write programs and execute them using external interpreters (Chen et al., 2022) to accurately yield the correct answer, as demonstrated in Appendix A.4. However, for more complex scenarios, it may still require further works such as designing sophisticated systems consisting of multiple AI Agents (Xi et al., 2023).

## 5 CONCLUSION

In this paper, we use a synthetic setting to demonstrate that LCLMs always struggle to solve multi-matching retrieval and logic-based retrieval, which are basic and common factors encompassed by those more advanced long-context tasks. Then, we find their underlying hyper-multi-step nature though more in-depth analysis, proving they must not be simply addressed by LLMs within limited steps. Therefore, we highlight that merely extending the context window size and relying solely on LCLMs to accurately address long-context tasks in various real-world scenarios may be futile. We urge more novel perspectives and approaches to further exploit long contexts.

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

# A APPENDIX: EXAMPLES OF PROMPTS AND MODEL RESPONSES

## A.1 PROMPT EXAMPLE OF KV RETRIEVAL

Here is an example of the context of a KV retrieval task and different types of the appended questions, including direct retrieval, formally multi-step retrieval, multi-matching retrieval, multi-matching retrieval but only need to retrieve the last one, and logic-based tasks:

---

**Context**

Json data with 3000 key-value pairs:
{"1532968704": 78, "5921306748": 84, "3742815096": 47, ......, "3276918540": 76}

---

**Question**

**Direct Retrieval:** In the above json data, please find the value of the key '6978024153'. Give your final answer (the value) in format of "value: {answer}"

- - - - - - - - - - - - - - - - - - - - - - - - - - - - - - - - - - - - - - - - - - - - -

**Reasoning and aggregation (multi-step):**
Question: In the above json data, please find the value (you need to search it in the Json dictionary) of the Key. The Key is the string S.
S is the sequential concatenation of A and B.
A is the sequential concatenation of the corresponding values (you need to search it in the Json dictionary) of the keys "8517603942", "2681307945", "3759160248", "6201843957", "7138095462". When concatenating, each value is seen as a character.
B is a string "85467".
Let's think step by step, and give your final answer (the key and the value) in format of "key:{answer} value:{answer}"

- - - - - - - - - - - - - - - - - - - - - - - - - - - - - - - - - - - - - - - - - - - - -

**Multi-matching retrieval:**
Question: In the above json data, please find all the keys with the value 0. Give your answer (the keys separated by comma) in format of "keys: {answer}"

- - - - - - - - - - - - - - - - - - - - - - - - - - - - - - - - - - - - - - - - - - - - -

**Multi-matching retrieval (only need to retrieve the last one):**
Question: In the above json data, please find all the keys with the value 0. You only need to find one more key to complete the answer. Answer: The 3 keys whose value is 0 are:
key 1: "1864209357"
key 2: "3145286970"
key 3: {answer}

- - - - - - - - - - - - - - - - - - - - - - - - - - - - - - - - - - - - - - - - - - - - -

**Logic based retrieval:**
Question: In the above json data, please find the Key (only one) whose Value (an integer) is greater than 223 and smaller than 278. Give your answer (the key) in format of "key: {answer}"

## A.2 PROMPT EXAMPLES OF STUDENT RESUME RETRIEVAL

Here is an example of the student resume retrieval task. All the students information are converted to declarative sentences of a fixed format in the context.

---

**Context**

Here are 100 students' resumes:

The student named Hallie Turner is 21 years old, graduated from New York University with a GPA of 4.96. He/She is interested in Social Media, Writing and his/her self-introduction is: Creative writer exploring the impact of social media on culture.

The student named Sonali Jain is 22 years old, graduated from Mithibai College with a GPA of 2.52. He/She is interested in Art, Writing, Travel and his/her self-introduction is: An artist and writer inspired by travels across cultures.

The student named ......

---

**Question**

**Simple retrieval:**
Question: What is the age of the student named Hallie Turner?

- - - - - - - - - - - - - - - - - - - - - - - - - - - - - - - - - - - - - - - - - -

**Multi-matching retrieval:**
Question: Please find all the students who graduated from Tokyo University of Agriculture and Technology. Please give your final answer (the students' names separated by commas) in the format of "names: {answer}"

- - - - - - - - - - - - - - - - - - - - - - - - - - - - - - - - - - - - - - - - - -

**Logic based retrieval:**
Question: Which student has a GPA between 2.25 and 2.58? Please give your final answer (the student's name) in the format of "name: {answer}"

---

## A.3 EXAMINE EACH ITEM ONE BY ONE TO RETRIEVE

Taking a logic-based KV retrieval task for example, when given a normal question without CoT-like prompt, the model (gpt-4o) tends to make mistakes:

---

**Context**

Json data with 100 key-value pairs:

{"0214587963": 933, "9578042316": 354, "8593746120": 143, "9473061852": 353, ...... "0587214936": 448, "6742193850": 379, **"2405163897": 572**, "4871369052": 768, ...... "6028917354": 448, "3249758610": 938}

---

**Standard Q&A**

**Question:** In the above json data, please find the Key (only one) whose Value (an integer) is greater than 527 and smaller than 620. Give your answer (the key) in format of "key: {answer}"

- - - - - - - - - - - - - - - - - - - - - - - - - - - - - - - - - - - - - - - - - -

**Wrong Answer:** The key whose value is greater than 527 and smaller than 620 is "1748263950" with a value of 525.

---

If we use a special CoT-like prompt to let it to examine each item one by one, it generates about 2000 tokens to judge every KV pair, which takes a long time but finally gives the right answer (for multi-matching problems, the process is similar):

---

**Check One By One Q&A**

**Question:** In the above json data, please find the Key (only one) whose Value (an integer) is greater than 527 and smaller than 620. Please check each item one by one and return your judgment (yes/no) on whether it meets the requirements. Give your final answer (the key) in format of "key: {answer}"

- - - - - - - - - - - - - - - - - - - - - - - - - - - - - - - - - - - - - - - - - - - - - - - - - - - -

**Answer:** To find the key whose value is greater than 527 and smaller than 620, let's examine each key-value pair in the provided JSON data:
1. "0214587963": 933 (no)
2. "9578042316": 354 (no)
......
50. "6742193850": 379 (no)
51. "2405163897": 572 (yes)
52. "4871369052": 768 (no)
......
100. "3249758610": 938 (no)

The key that meets the requirement is "2405163897" with a value of 572.
Final answer: key: 2405163897

---

## A.4 USE PROGRAM OF THOUGHTS TO RETRIEVE

---

**Program of Thoughts Q&A**

**Question:** In the above json data, please find the Key (only one) whose Value (an integer) is greater than 527 and smaller than 620. You can write a python program to solve this, and regard the json data as a file "KV.json".

**Answer:** To find the key whose value is greater than 527 and smaller than 620, you can use the following Python program. This program reads the JSON data from a file named "KV.json" and searches for the key that meets the criteria.

```python
import json

# Load the JSON data from the file
with open("KV.json", "r") as file:
    data = json.load(file)

# Initialize a variable to store the desired key
desired_key \= None

# Iterate over the key-value pairs in the JSON data
for key, value in data.items():
    # Check if the value is greater than 527 and smaller than 620
    if 527 < value < 620:
        desired_key = key
        break # Since we only need one key, we can break once we
            find it

# Print the desired key
print("The key with value greater than 527 and smaller than 620
    is:", desired_key)
```

This program will output the key that satisfies the condition. When you run it with the provided JSON data, it will find and print the key '"2405163897"', which has the value '572'.

---

We can also use program-of-thoughts prompt (Chen et al., 2022) to let the model write programs to solve this retrieval task. Writing python program to solve this task is very easy for gpt-4o, and then by executing this program in an external interpreter, we can get the right result. Here is an example

---

of logic-based KV retrieval (it can similarly easily solve multi-matching problems), whose context is the same as the previous section.

## A.5 ARITHMETIC PROBLEM

Finding the biggest number among $N$ numbers typically needs $N$ steps, which belongs to the simplest multi-step logical problems. Our prompt is as the following, where the model will predict the correct answer directly after the prompt.

> A list of integers: 15, 24, 31, 44. In the list, the biggest integer is

## A.6 SENTENCE EMBEDDING MODELS ON MATH PROBLEMS

We test 2 commonly used sentence embedding model, e5-large-multilingual (Wang et al., 2024a) and bge-m3(Zhang et al., 2023), on a numerical value comparing task to see if it can retrieve the correct number from 20 integers. The candidate keys are integers within the range 0 to 30, 100, 1,000, or 10,000, and the queries have 2 types: (1) equality judgement (2) greater&less than judgement. As the example:

---
**Query and Candidates**

**query (equal)**: The integer equal to 310.

- - - - - - - - - - - - - - - - - - - - - - - - - - - - - - - - - - - - - - - - -

**query (greater than and less than)**: The integer smaller than 223 and larger than 356.

- - - - - - - - - - - - - - - - - - - - - - - - - - - - - - - - - - - - - - - - -

**20 candidates (range 0-1000)**: 310, 734, 296, 501, 893, 178, 645, 923, 57, 782, 464, 852, 213, 689, 371, 970, 510, 116, 455, 682

---

## A.7 ANCHOR TOKEN SELECTION

The selected anchor tokens in different question types are highlighted in red in the examples below:

---
**Anchor Tokens**

**Direct retrieval:** {Context} In the above JSON data, the Key whose Value is 5 is: "

**Logic-based retrieval:** {Context} In the above JSON data, the Key whose Value is larger than 4 and smaller than 6 is: "

**Multi-matching retrieval:** {Context} In the above JSON data, all the Keys whose Value is 5 are: "1532968704", "5921306748", "3742815096"

---

# B APPENDIX: ADDITIONAL EXPERIMENTS

## B.1 DIRECT RETRIEVAL: USUALLY SIMPLE, BUT SENSITIVE TO QUESTION TYPE

Previous benchmarks have demonstrated the model's strong performance in direct retrieval tasks such as NIAH (gkamradt, 2023) and Key-Value retrieval Zhang et al. (2024), even when the context length exceeds 100k tokens. However, interestingly, our experiments with 2 simple types of direct retrieval tasks with only slight differences reveal that some models may be more sensitive to the question type, while the extremely long context itself is not of primary concern.

For KV retrieval task across different context lengths, we test two types of problems: searching for a value given a key (k-v), and searching for a key given a value (v-k). The results, shown in Table 6, indicate that in the "k-v" task, increased context length does not significantly affect model performance, because the task is inherently simple for the model. However, for phi-3.5 and

deepseek, a slight modification in the problem to a "v-k" form, while keeping the context length constant, leads to a sharp decline in performance, which is much greater than that caused by simply increasing the context length for the same problem. This suggests that models are more sensitive to the type of problem than to the length of the context. (We infer that models like phi-3.5 and deepseek may treat "v-k" task as a numeric comparing task rather than direct retrieval, thus perform very poor in long context.)

Therefore, we conclude, current long-context models can indeed perfectly pass tests like Key-Value retrieval or NIAH (gkamradt, 2023) in 128k length, but they may be highly susceptible to the specific form of questions, even when the context itself remains unchanged. So we speculate that seemingly similar forms of problems may have fundamentally different natures.

| Task | Model | $N=10$ | $N=100$ | $N=1000$ | $N=3000$ |
|------|-------|--------|---------|----------|----------|
| k-v | phi-3.5 | 100 | 99 | 92 | 85 |
|     | llama3.1 70b | 100 | 100 | 99 | 100 |
|     | deepseek | 100 | 100 | 100 | 100 |
|     | gpt-4o | 100 | 100 | 100 | 100 |
| v-k | phi-3.5 | 98 | 67 | 7 | 0 |
|     | llama3.1 70b | 100 | 100 | 99 | 100 |
|     | deepseek | 100 | 99 | 10 | 0 |
|     | gpt-4o | 100 | 100 | 100 | 100 |

Table 6: model score on simple KV retrieval tasks with 2 problem types: k-v means given key, search value and v-k means given value, search key. The context lengths with respect to KV number 10, 100, 1000, 3000 are 0.1k,1k,10k,30k.

### B.2 MULTI-STEP RETRIEVAL: A LITTLE HARDER BUT CAN BE SOLVED BY COT

Multi-step (in form) retrieval tasks, including multi-query, multi-hop, chain-of-retrieval tasks, etc. (Li et al., 2024; Wang et al., 2024b), intuitively appear more challenging due to the need to aggregate dispersed information from the context by multiple steps. Nevertheless, our experiments indicate that they are not necessarily difficult, provided LLMs can decompose them into simpler steps using CoT approach (Wei et al., 2022). We must emphasize that, it is important to distinguish multi-matching from formally multi-step retrieval: the former requires retrieving multiple items with only one query, while the latter also requires retrieving multiple items but can be achieved with multiple different queries.

Previous studies (Wang et al., 2024b; Goldman et al., 2024; Li et al., 2024; 2023) typically characterize long-context tasks, which require the aggregation of dispersed information for multi-step reasoning, as difficult. While this assessment aligns with intuition, we argue that this classification is not universally applicable. Instead, the complexity of such tasks should be evaluated based on their specific form and composition, because a complex task may not be inherently difficult if a model can systematically reason and decompose the task into manageable steps.

To illustrate this, we constructed a multi-step Key-Value retrieval problem that involves a chain-of-retrieval process. This task requires a model to retrieve $n$ values corresponding to $n$ keys, concatenate them into a string, and then combine this string with another given string to generate a new key, from which the model must retrieve its value as the final answer (see detailed prompts in Appendix A.1). Although this problem seems complex, requiring aggregating at least $n+1$ pieces of information from different parts of the context and performing at least $n+1$ reasoning steps, it can actually be decomposed into simple steps using a chain-of-thought (CoT) approach (Wei et al., 2022).

The results presented in Table 7 reveal that smaller models, such as phi-3.5, perform worse as the number of reasoning steps and pieces of information increase. However, larger models with stronger reasoning capabilities are almost unaffected. This suggests that while long-context problems which must require multi-step may pose challenges for smaller models, they are not unsolvable for larger models with robust reasoning and comprehension skills. In other words, multi-step problems primarily test a language model's comprehension, reasoning, and organizational abilities, rather than its retrieval skills or other capabilities specifically related to handling long contexts.

| Model | Total KVs ($N$) | 1 step | 3 steps | 5 steps |
|---|---|---|---|---|
| phi-3.5 | 10 | 85, 68 | 58, 49 | 15, 17 |
| | 100 | 60, 48 | 57, 30 | 3, 13 |
| | 1000 | 62, 10 | 13, 0 | 0, 7 |
| llama3.1 70b | 10 | 99, 95 | 97, 95 | 78, 83 |
| | 100 | 96, 96 | 97, 92 | 71, 78 |
| | 1000 | 96, 88 | 91, 55 | 50, 50 |
| deepseek | 10 | 100, 100 | 100, 100 | 100, 100 |
| | 100 | 100, 93 | 100, 100 | 100, 100 |
| | 1000 | 100, 40 | 97,40 | 93,60 |
| gpt-4o | 10 | 100, 100 | 100, 100 | 100, 100 |
| | 100 | 100, 100 | 100, 100 | 100, 100 |
| | 1000 | 100, 100 | 100, 100 | 94, 97 |

Table 7: Model performance on tasks requiring gathering $n$ values to form a new key and then retrieve its value. We record 2 scores, the former is the accuracy of forming the Key and the latter is the accuracy of getting the Value.

## B.3 DETAINED EVALUATION RESULTS OF MULTI-MATCHING RETRIEVAL

| Model | Total KVs | 1 match | 5 matches | 10 matches | 20 matches |
|---|---|---|---|---|---|
| phi-3.5 | 10 | 91 (7/0/2) | 36 (5/30/29) | / | / |
| | 100 | 42 (35/2/21) | 0 (0/10/90) | 0 (0/0/100) | 0 (0/0/100) |
| | 1000 | 1 (7/13/79) | 1 (7/0/87) | 0 (0/7/93) | 0 (0/0/100) |
| llama3.1 70b | 10 | 100 (0/0/0) | 99 (1/0/0) | / | / |
| | 100 | 99 (0/0/0) | 59 (17/14/8) | 45 (14/17/22) | 24 (12/34/30) |
| | 1000 | 62 (0/0/0) | 3 (2/47/46) | 0 (0/15/85) | 0 (0/12/88) |
| deepseek | 10 | 100 (0/0/0) | 100 (0/0/0) | / | / |
| | 100 | 95 (3/0/2) | 40 (9/36/15) | 13 (9/42/36) | 3 (3/43/51) |
| | 1000 | 0 (0/100/0) | 0 (0/100/0) | 0 (0/100/0) | 0 (0/100/0) |
| gemini | 10 | 100 (0/0/0) | 100 (0/0/0) | / | / |
| | 100 | 100 (0/0/0) | 82 (14/1/3) | 48 (32/12/8) | 35 (14/36/15) |
| | 1000 | 94 (2/4/0) | 66 (3/26/5) | 18 (9/47/26) | 2 (4/45/49) |
| gpt-4o | 10 | 100 (0/0/0) | 100 (0/0/0) | / | / |
| | 100 | 100 (0/0/0) | 98 (0/2/0) | 85 (5/5/5) | 50 (5/40/5) |
| | 1000 | 60 (5/35/0) | 45 (5/45/5) | 5 (0/50/45) | 0 (0/17/83) |

Table 8: Accuracy on KV retrieval, requiring retrieve all the keys with the given value, when the number of total KVs and matching KVs are increasing.

For multi-matching retrieval, the model generates a list of items as the answer, which can be categorized into four distinct cases:

1. **Fully Correct**: The model's response exactly matches the correct answer, i.e., both sets are equal.

2. **Over-selection**: The correct answer is a proper subset of the model's response.

3. **Under-selection**: The model's response is a proper subset of the correct answer.

4. **Mis-selection**: The model's response and the correct answer do not overlap as subsets of each other.

| Model | Total Students | 1 match | 5 matches | 10 matches |
|---|---|---|---|---|
| phi-3.5 | 10 | 98 (1/0/1) | 27 (13/47/13) | 98 (0/2/0) |
| | 100 | 33 (52/0/15) | 0 (1/4/95) | 0 (0/1/99) |
| llama3.1 70b | 10 | 100 (0/0/0) | 99 (0/1/0) | 100 (0/0/0) |
| | 100 | 98 (1/0/1) | 21 (8/48/23) | 0 (3/45/52) |
| deepseek | 10 | 100 (0/0/0) | 91 (1/8/0) | 100 (0/0/0) |
| | 100 | 100 (0/0/0) | 23 (1/54/22) | 0 (1/36/63) |
| gemini | 10 | 100 (0/0/0) | 100 (0/0/0) | 100 (0/0/0) |
| | 100 | 96 (4/0/0) | 36 (8/47/9) | 8 (5/37/50) |
| gpt-4o | 10 | 100 (0/0/0) | 100 (0/0/0) | 100 (0/0/0) |
| | 100 | 100 (0/0/0) | 65 (0/20/15) | 0 (0/75/25) |

Table 9: Accuracy on student resume retrieval, requiring retrieve all the students graduating from the given university, when the number of total students and matching students vary.

In Table 8 and Table 9, we detail the model's performance, with the first number representing the rate of fully correct responses, while the numbers in parentheses denote over-selection, under-selection, and mis-selection, respectively.

### B.4 DETAILED ATTENTION WEIGHTS

In Figure 5 we show the absolute attention, i.e., the original attention weights to the gold Key or Value in KV retrieval tasks.

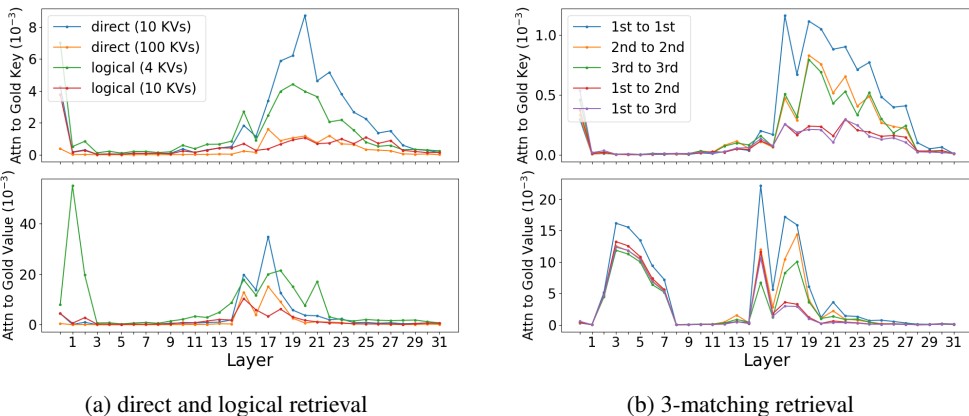

(a) direct and logical retrieval    (b) 3-matching retrieval

Figure 5: (a)Attention to each Key or Value in layer 17 when predicting the first, second and third key respectively. We highlighted the bar corresponding to the matching Keys in red.

In Figure 6, we illustrate the average attention distributed to each Key or Value in the context in layer 17 (because the phenomenons in layer 17 to 21 are similar to that in layer 17) of phi-3.5, in a 3-matching KV retrieval task, when predicting the 3 Keys respectively (the query tokens we choose are shown in section 4.1). Interestingly, the model actually has paid much attention to all the 3 matching KVs at first, but the attention when predicting the 2nd or 3rd seems to be more dispersed to unrelated KVs though it has actually found the correct answer to some extent, which may introduce noise to harm the accurate retrieval.

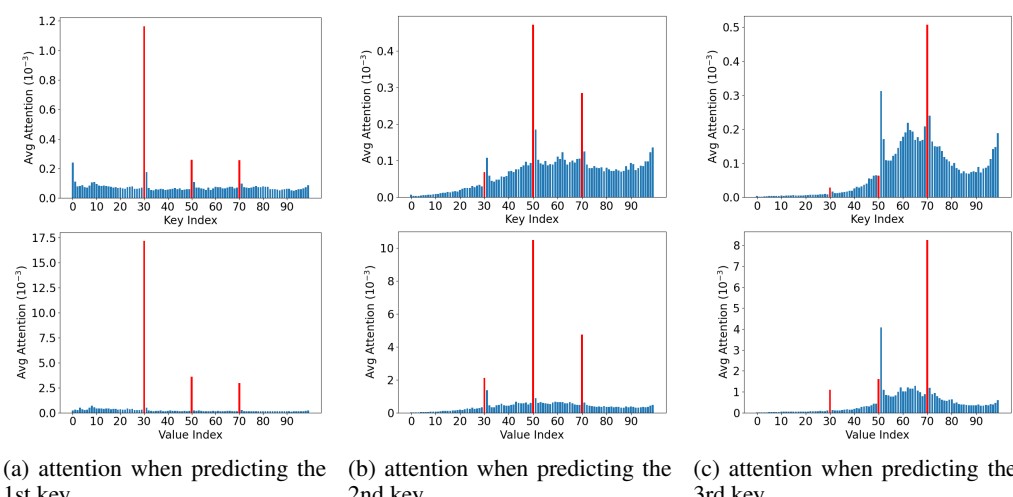

(a) attention when predicting the 1st key

(b) attention when predicting the 2nd key

(c) attention when predicting the 3rd key

Figure 6: Attention to each Key or Value in layer 17 when predicting the first, second and third key respectively. We highlighted the bar corresponding to the matching Keys in red.

## B.5 RETRIEVING JUST ONE ITEM IN MULTI-MATCHING RETRIEVAL

The full results of models' performance when asked to retrieve just the last item in in multi-matching KV retrieval is shown in Table 10.

| Model | Total KVs | 3 matches | 10 matches | 30 matches | 100 matches |
|---|---|---|---|---|---|
| phi-3.5 | 100 | 22 | 2 | 0 | / |
|  | 1000 | 0 | 0 | 0 | 0 |
| llama3.1 70b | 100 | 91 | 66 | 54 | / |
|  | 1000 | 61 | 43 | 16 | 1 |
| deepseek | 100 | 93 | 75 | 37 | / |
|  | 1000 | 15 | 5 | 6 | 0 |
| gpt-4o | 100 | 100 | 92 | 42 | / |
|  | 1000 | 95 | 42 | 25 | 1 |
| gemini | 100 | 100 | 99 | 67 | / |
|  | 1000 | 99 | 90 | 49 | 8 |

Table 10: Accuracy on KV retrieval, requiring retrieving only the last Key with the given Value, and other Keys with the given Value are already given.

## C    ANALYSE THE COMPONENTS OF CHALLENGING TASKS FROM VARIOUS LONG-CONTEXT BENCHMARKS

### C.1    DISTINGUISHING WHAT IS MULTI-MATCHING OR LOGIC-BASED RETRIEVAL

We regard multi-matching and logic-based retrieval as important individual components because of their distinct natures. It may be difficult to distinguish between multi-matching and multi-query, or logic-based retrieval and problems requiring logic. Here we use some simple example to illustrate this. The primary characteristic of multi-matching or logic-based retrieval is the inability to be further divided into multiple manageable steps using CoT, unless examining each item in the context one by one.

Multi-matching means multiple items meet one retrieval criteria, which can almost no longer be subdivided into multiple independent conditions. However, as for multi-query, although the retrieval criteria is also in one sentence, it can be easily subdivided.

> **Context**
>
> * Jack is 30 years old.
> * Mike is 20 years old.
> * Lee is 50 years old.
> * William is 20 years old.
> * James is 35 years old.

Here is the example, where the multi-query retrieval question can be subdivided into 3 questions to retrieve the 3 people individually, while the multi-matching retrieval problem cannot.

> **Question Differentiation (1)**
>
> **Multi-query retrieval (easy):** How old is Jack, Lee and James respectively?
>
> **CoT**: (1) retrieve Jack's age (2) retrieve Lee's age (3) retrieve James's age (4) summary
>
> - - - - - - - - - - - - - - - - - - - - - - - - - - - - - - - - - - - - - - - -
>
> **Multi-matching retrieval (hard):** Who are 20 years old ?

Here is the example, where a problem requiring logical reasoning can be easily devided into simple 4 steps, while the logic-based retrieval problem cannot.

In our study, "logic" typically specifically refers to advanced logical judgments, such as determining that 50 is greater than 45. Although "equality" is also a logical relationship in mathematics, the equivalence of two identical numbers can always be established through low-level abstract similarity alone. Therefore, LLMs or embedding models do not require advanced "logic" to judge equality relationships. Therefore, retrieval based on equality is not considered a logic-based retrieval problem in our study.

> **Question Differentiation (2)**
>
> **Requiring logical reasoning (easy):** Whose age is equal to the age of Lee minus the age of Jack?
>
> **CoT**: (1) retrieve Lee's age (2) retrieve Jack's age (3) do a simple subtraction and get the target age (4) retrieve by the target age
>
> - - - - - - - - - - - - - - - - - - - - - - - - - - - - - - - - - - - - - - - -
>
> **Logic-based retrieval (hard):** Who is older than 45?

## C.2 ANALYSE ADVANCED LONG-CONTEXT TASKS

In Table 11, we list some advance long-context tasks from previous benchmarks, and identify if each task is very hard, is multi-step in form, involves multi-matching retrieval or logic-based retrieval.

If the evaluation results from this benchmark indicate that no existing model is able to score above 60 on this task, we will categorize the task as very difficult; otherwise, it will be marked as not very difficult. How we decide whether it is multi-step, multi-matching or logic-based is shown in Appendix C.3.

Note that multi-matching will not be very difficult if the amount of matching items is not so large, for example, a 3-matching retrieval may be easy for Gemini, but 30-matching must be really hard.

| Benchmark | Task Name | hard | multi-step | multi-match | logic-based |
|---|---|---|---|---|---|
| Loogle | Multiple info retrieval | ✓ | ✗ | ✓ | ✗ |
| | Computation | ✓ | ✓ | ✗ | ? |
| | Timeline reorder | ✓ | ✓ | ✗ | ✓ |
| | Comprehension&reasoning | ✓ | ✓ | ? | ? |
| Ruler | Multi-keys NIAH | ✗ | ✗ | ✗ | ✗ |
| | Multi-values NIAH | ✗ | ✗ | ✓ | ✗ |
| | Multi-queries NIAH | ✗ | ✓ | ✗ | ✗ |
| | Multi-hop Tracing | ✗ | ✓ | ✗ | ✗ |
| | Aggregation | ✗ | ✓ | ✓ | ✗ |
| NeedleBench | Multi-Needle Retrieval | ✗ | ✓ | ✗ | ✗ |
| | Multi-Needle Reasoning | ✗ | ✓ | ✗ | ✗ |
| | Ancestral Trace | ✗ | ✓ | ✗ | ✗ |
| Loong | Spotlight Locating | ✗ | ✗ | ✗ | ✗ |
| | Comparison | ✓ | ✗ | ? | ✓ |
| | Clustering | ✓ | ✗ | ? | ✓ |
| | Chain of Reasoning | ✗ | ✓ | ? | ? |

Table 11: Some advanced long context tasks from different benchmarks. We mark whether the task is very hard, is multi-step in form, involves multi-matching retrieval or logic-based retrieval. ? means this is uncertain, depending on more specific scenarios.

### C.3 DETAILS OF THESE TASKS

Here we show the process that we analyse previous challenging tasks from different benchmarks to identify which component these tasks involve. We omit simple tasks like single-needle NIAH or normal multi-document QA.

#### C.3.1 LOOGLE

Loogle (Li et al., 2023) is a long-context benchmark which first distinguish short-dependency tasks and long-dependency tasks. Long-dependency means the task needs a large portion of the context rather than just a short part, which is much more challenging. We analyse the 4 task types belonging to long-dependency tasks:

**Multiple information retrieval:** This task is quite different from traditional short-term retrieval tasks, there are usually multiple and diverse pieces of evidence throughout the entire text for one specific answer. This task usually involves multi-matching retrieval, but sometimes can also be separable into steps.

**Computation:** This task firstly needs multiple information retrieval from a wide range of texts, and then use these data for calculating. A majority of the evidence within the text takes the form of numerical data. However, this task is in fact composed of 3 solvable steps: understand the question to determine which numeric to retrieve, get the numeric through normal retrieval operation (step 1 and 2 may be perform several times to get multiple numeric), calculate the answer based on the retrieved data. Thus it does not belong to logical retrieval, but may sometimes involve multi-matching.

**Timeline reorder:** This task requires reordering the timeline of a set of events presented in a permuted order. It apparently needs to compare the size of numbers to determine which event should be retrieved first or later. So it involves logic-based retrieval.

**Comprehension and reasoning:** This task demands not only a profound comprehension of the question but also intricate reasoning to discern the underlying implications for searching for the appropriate evidence. It must be a multi-step problem, but whether this issue involves other components remains uncertain and depends on the specific nature of the problem in question.

### C.3.2 RULER

We choose 5 difficult tasks from Ruler (Hsieh et al., 2024) to analyse:

**Multi-keys NIAH:** Multiple "needles" are inserted into the "haystack", and only one of them needs to be retrieved. The additional "needles" are hard distractors. This is a normal retrieval task. Though many hard distractors are inserted, powerful LLMs can usually overcome this.

**Multi-values NIAH:** Multiple "needles" sharing the same key are inserted into the "haystack". All values associated with the same key need to be retrieved. This is totally the same as the multi-matching retrieval.

**Multi-queries NIAH:** Multiple "needles" with distinct keys are inserted into the "haystack". On the surface, it may appear that a problem requires the retrieval of multiple values. However, since each value has a distinct key, it can actually be decomposed into multiple simple retrieval tasks. Therefore, this does not fall under the category of logic-based retrieval or multi-match retrieval.

**Multi-hop Tracing:** A variable X1 is initialized with a value V, followed by a linear chain of variable name binding statements (e.g., X2 = X1, X3 = X2, ...), which are inserted at various positions of the input. The objective is to return all variable names pointing to the same value V. This is a classic chain-of-retrieval task. It can also be decomposed into multiple simple retrieval tasks, e.g., first retrieve X1, then use X1 as the query to retrieve X2. Therefore, it is multi-step, but not multi-matching. It may be logic-based retrieval if the variable name binding statements involve more complex calculations.

**Aggregation:** This task includes Common Words (CWE) and Frequent Words Extraction (FWE). A model needs to return the top-K frequent words in the context. This is a very hard multi-step problem consisting of at least 3 steps: identify each word, use each word as the query to do multi-matching retrieval and compare the frequency of each word. So it must involve multi-matching.

### C.3.3 NEEDLEBENCH

NeedleBench (Li et al., 2024) aims to let NIAH gkamradt (2023) more challenging. We analyse all the tasks from it, except the simplest one, Single-Needle Retrieval Task.

**Multi-Needle Retrieval Task:** This task is nearly the same as Multi-queries NIAH. It can actually be decomposed into multiple independent retrieval problems, thus it is not difficult.

**Multi-Needle Reasoning:** In this task, the model must first engage in reasoning to comprehend the issue, thereby determining which specific pieces of information are required. Subsequently, it must retrieve these multiple pieces of information from the context. This task is also multi-step, which can be solved by CoT (Wei et al., 2022). However, none of the steps necessitates logic-based or multi-matching retrieval.

**Ancestral Trace Challenge:** The context encompasses a multitude of interpersonal relationships, and the model is tasked with discerning the ancestral relationship between two individuals. This is similar to Multi-hop Tracing, so it does not necessitate logic-based or multi-matching retrieval.

### C.3.4 LOONG

Loong (Wang et al., 2024b) is a recent long-context benchmark which emphasized the challenge of the task. Most tasks in it involves mathematical calculation, which is very hard for LLMs. We analyse all of the 4 tasks from it.

**Spotlight Locating:** It is aimed at examining the LLMs' ability to search the evidence within one document from multiple ones. So it is a simple retrieval task.

**Comparison:**    One of the sub-tasks is that given a specific numerical or conceptual range, the model should output all objects within multiple documents that meet the condition. Apparently, this task is a typical logic-based retrieval task.

**Clustering:**    One of the sub-tasks requires the model to group the evidence existing in the provided financial reports into corresponding sets based on textual or numerical criteria. Apparently, this task is a typical logic-based retrieval task, too.

**Chain of Reasoning:**    This task evaluates the model's proficiency in logical reasoning, which requires LLMs to locate the corresponding evidence within multiple documents and model the logical relationships among them for deducing the answer. This is similar to Multi-hop Tracing, which is a multi-step reasoning task, but whether involving logic-retrieval or multi-matching should depends on specific problems.

