# OpenReview forum: "Hyper-multi-step: The Truth Behind Difficult Long-context Tasks"
_ICLR.cc/2025/Conference — ICLR 2025 Conference Withdrawn Submission_

### Official Review · Reviewer_r8ab · 2024-10-31

**Soundness:** 3
**Presentation:** 2
**Contribution:** 2
**Rating:** 3
**Confidence:** 4

**Summary:**

This paper empirically analyzed how long-context LLMs perform on multi-matching retrieval and numerical comparison tasks. Synthetic test cases were designed and used. Authors attributed the difficulty to "hyper-multi-step", i.e., numerous steps to solve.

**Strengths:**

S1. Analyzing the behavior of long-context LLMs is interesting and timely.

S2. Implementation details are sufficient.

**Weaknesses:**

W1. Experiments can be better designed.

First, for multi-matching retrieval, it is not fair to use totally exact match as the evaluation metric. Why not using F1 score which can more fine-grainedly reflect model performance?

Second, I am confused by the difference in prompts in Section 3.1 and in Section 4.1. Why not using the prompts in Section 4.1, e.g., those in Appendix A.7, to conduct your main experiments in Section 3.1? Given this difference in prompts, is it possible that your analysis in Section 4 cannot align with your results in Section 3?

W2. Results are generally trivial.

The main conclusions, such as "model behavior in logic-based retrieval ... more resembles the numeric comparison task", "one-step vector retrieval techniques are inadequate for logic-based retrieval", and "predicting later items is increasingly harder for LLMs", are trivial and already given in the literature. I did not see new findings throughout the paper. The concept of "hyper-multi-step" is also trivial.

**Questions:**

See my questions in Weaknesses.

---

### Official Review · Reviewer_4Xep · 2024-11-02

**Soundness:** 2
**Presentation:** 1
**Contribution:** 2
**Rating:** 3
**Confidence:** 4

**Summary:**

This work examines why long-context language models struggle with certain complex tasks. The authors provide two simple retrieval tasks: kv retrieval and student resume retrieval. Subsequent experiments are conducted on 5 popular LLMs. They assume these tasks are hyper-multi-step, requiring more steps than models can handle.

**Strengths:**

- The authors provide detailed explanation about the two tasks.
- The experiments are easy to follow.
- The experimental results align with the readers' intuition.

**Weaknesses:**

- The construction of the tested tasks does not align with the authors' motivation to  to reveal the real source of challenges in long context tasks. The two tasks presented in this work are overly simplistic and fail to represent the complex inputs that LCLMs encounter in real applications. Excelling at identifying students who graduated from a specific university does not seem to contribute significantly to real-world applications like long-context summarization.
- These two tasks lack novelty: KV Retrieval is a common retrieval task used in much previous work, such as InfiniteBench [1]. Both KV Retrieval and Student Resume Retrieval can be classified as "Needle-in-a-Haystack" tasks, which aim to find specific information within a long noisy context. However, diverse works like LongIns [2] and NOCHA [3] challenge this setting, arguing that it is synthetic and does not reflect performance in real-world applications.
- The settings are inappropriate. The experiments and analysis in Section 4 do not explain LLMs' behavior in long contexts, as the input is simplified into a short context. Additionally, the lack of a chat template does not align with the paradigm of instruction-tuned LLMs. Therefore, I do not believe the analysis can be used to improve long-context modeling.

References:
1. InfiniteBench: Extending Long Context Evaluation Beyond 100K Tokens.
2. LongIns: A Challenging Long-context Instruction-based Exam for LLMs.
3. One Thousand and One Pairs: A "Novel" Challenge for Long-Context Language Models.

**Questions:**

- I am a bit confused about the inference setting in Section 3. Why do the authors not use greedy decoding for these retrieval tasks? Additionally, I don't think the maximum of 512 generated tokens may are sufficient when the number of matches exceeds to 10. LLMs often reason before providing answers, even without a cot requirement in the prompts.
- I find this work disjointed. The first three sections focus on long context, while Section 4 shifts to the retrieval and reasoning behavior of LLMs with short context. The authors should reconsider the coherence of their narrative and presentation.

---

### Official Review · Reviewer_Y9Rb · 2024-11-03

**Soundness:** 3
**Presentation:** 2
**Contribution:** 2
**Rating:** 3
**Confidence:** 4

**Summary:**

The authors investigate the underlying causes of weak performance of LLMs on long-context benchmarks, They identify multi-match and logic-based retrieval as main causes, and perform experiments aimed to uncover the extend to which these can / cannot be solved.

**Strengths:**

- understanding performance on benchmarks by analysing the causes of difficulty is certainly a good idea
- experiments are done well

**Weaknesses:**

- analysis of benchmarks is hidden in appendix C, and it is unclear how complete this analysis is (a more complete analysis and discussion should have been part of the paper)
- all experiments are done on two synthetic datasets / tasks, but it remains unclear how representative these are for the benchmarks
- some experiments are redundant, i.e. it is obvious that vector-based retrieval cannot handle more advanced logical operations such as greater-than or less-than comparison - you would need a database query language like SQL or similar to perform these queries well
- what is missing is a discussion of how these difficult tasks could possibly be solved (see the comment above)

**Questions:**

1. How representative are your two synthetic datasets / tasks for the difficult tasks contained in the benchmarks?
2. On your datasets, to perform them well, a database query language like SQL would likely perform well. Can you comment?

---

### Official Review · Reviewer_1XWZ · 2024-11-11

**Soundness:** 2
**Presentation:** 2
**Contribution:** 2
**Rating:** 5
**Confidence:** 4

**Summary:**

The paper conducts experimental analysis of the long context large language models on the task of multi-matching retrieval and logic based retrieval. These tasks are then further evaluated over the benchmark datasets and provides potential solutions after the experimental analysis.

**Strengths:**

--> The authors analyse the LCLMs such as GPT-4o and Gemini over the tasks requiring longer context windows. The tasks include multimatching and logic based retrieval.
--> Thorough experimentation has been performed and the paper is written very well.

**Weaknesses:**

--> The main outcomes of the analysis should be emphasized upon in the beginning.
--> Why were the benchmarks listed in section 2.2 were not used for experimental comparison.
--> Most of the time the analysis was referred to in the Appendix of the paper.
-- > Section 3.1 described the dataset is very much detail, it is not clear if the dataset is created by the authors or is it already existing. If this is a contribution then it should be highlighted in reference to existing work.
--> The related work was simply dumped and not compared with.
--> A deeper analysis giving the examples of where the models fail would also be interesting to see in details (something close to an error analysis).
--> The research contribution is unclear.

**Questions:**

--> Why the authors selected these particular benchmarks were used?
--> The appendix contains a lot of interesting analysis also by implementing some of the possible solutions which would be interesting to have as the part of actual paper.

---

### Note · Authors · 2024-11-20

I have read and agree with the venue's withdrawal policy on behalf of myself and my co-authors.